# *Salmonella* Inactivation Model by UV-C Light Treatment in Chicken Breast

**DOI:** 10.3390/microorganisms12091805

**Published:** 2024-08-31

**Authors:** Rosa María García-Gimeno, Eva Palomo-Manzano, Guiomar Denisse Posada-Izquierdo

**Affiliations:** Department of Food Science and Technology, Unidad de Investigación Competitiva de Zoonosis y Enfermedades Emergentes (UIC ENZOEM), International Campus of Excellence in the AgriFood Sector (CeiA3), Universidad de Córdoba, 14014 Córdoba, Spain; t52pamae@uco.es (E.P.-M.); bt2poizg@uco.es (G.D.P.-I.)

**Keywords:** non-thermal treatment, UV-C, low transmittance food, mathematical modeling, foodborne pathogens

## Abstract

This study aims to evaluate the effectiveness of inactivating *Salmonella enteritidis* in fresh chicken breast by irradiation using a combination of short-wave UV (0, 3, 6, 9, 12, and 15 J/cm^2^) and a natural antimicrobial such as caffeine (0, 5, 10, 15, and 20 nM/g) at 14 °C as alternative proposals to conventional techniques to reduce pathogens in food. The effect of temperature was studied in an initial phase (2 to 22 °C). The most suitable models were double Weibull in 60% of cases, with an adjustment of R^2^ 0.9903–0.9553, and Weibull + tail in 46.67%, with an adjustment of R^2^ of 0.9998–0.9981. The most effective combination for the reduction in *Salmonella* was 12 J/cm^2^ of UV light and 15 nM/g of caffeine, with a reduction of 6 CFU/g and an inactivation rate of 0.72. The synergistic effect was observed by increasing caffeine and UV light. Furthermore, the physico-chemical characteristics of the food matrix were not affected by the combination of both technologies. Therefore, these results suggest that this combination can be used in the food industry to effectively inactivate *Salmonella enteritidis* without deteriorating product quality.

## 1. Introduction

The analysis of meat and meat products is a significant activity in the area of food safety and nutrition, since they represent an important and relatively broad component of the diet. Meat and meat products are covered by legislative requirements around the world [1]. In Europe, Regulation No. 853/2004 [2] is responsible for defining the conditions of legislation on meat and edible parts of animals, including blood. On the other hand, a meat product is referred to as a transformed product resulting from the transformation of the meat or from the new transformation of said transformed product so that the cut surface shows that the product has ceased to possess the characteristics of the fresh meat. Chicken breast is a very popular product because it is relatively inexpensive, has high nutritional value, and has many health benefits [3]. For this reason, a multitude of studies on the microbiological safety of this type of product have been carried out [4] during the manufacturing and distribution phase, which involves cutting, peeling, joining, and branching [5].

Salmonellosis is a gastrointestinal disease caused by ingestion of food contaminated with *Salmonella* or by handling animals or animal products contaminated with *Salmonella*. Currently, it remains the second most commonly reported gastrointestinal infection in humans after campylobacteriosis and a major cause of foodborne outbreaks in the EU [6]; therefore, it is considered a public health problem. 

*Salmonella* can enter the food supply through various routes, such as fecal contamination from food handlers. Food-producing animals such as chickens, pigs, and cows harbor *Salmonella* serotypes that are human pathogens and can be passed to people through fresh foods such as eggs, meat, and dairy products. Foodborne *Salmonella* infections often start from products such as custards, cream pies, meringues, pies, and eggnog made with raw eggs. Other foods that tend to be implicated in salmonellosis outbreaks are meats and meat products, especially poultry; raw cured sausages; and other meat, milk, and dairy products [6].

Consequently, a variety of interventions based on chemical, physical, or biological agents have been developed to prevent bacterial contamination of foods [7,8,9]. Among the physical agents, there are traditional treatments such as thermal ones, and the current trend in the food industry is the implementation of non-thermal treatments, which is driven by the strong preference of consumers for fresh and minimally processed foods. These technologies include pulsed electric fields, UV light processing, minimal thermal processes, high-pressure batch or continuous processing, and the use of natural antimicrobials, among many others [10].

The technology used in this work was short-wave UV radiation, which has numerous advantages, including the ability to inactivate a wide range of pathogenic microorganisms [11,12], thus minimizing the loss of nutritional and sensory quality [13]. Among the main advantages is the absence of chemical residues or toxic compounds during treatment [10]. The microbicidal properties of short-wave UV light depend on the absorption of DNA from UV light, which induces distortions in the DNA molecule, inhibiting transcription and replication and eventually leading to cell death [14]. The application of this technology has been successfully applied as a non-thermal method for food decontamination [15,16,17,18].

Caffeine is a plant alkaloid in plants such as coffee, tea, and cocoa [19], and it is a biologically active molecule used in the food and pharmaceutical industry. Also known by the IUPAC as 1,3,7-trimethyl-1H-purine-2,6(3H,7H)-dione, it is a purine alkaloid that contains pharmacological properties and a therapeutic agent with analeptic activity [20]. Some studies have shown that caffeine can significantly decrease the survival of *E. coli* strains grown in laboratory media [21], chicken breast [22], and gram-negative bacteria [23].

Malettab and Were (2012) performed a study using caffeine in chicken breasts as a dietary matrix. They concluded that coffee filtering did not contribute significantly to the antimicrobial effect on chicken breasts, although there was a correlation between the presence of coffee filtrate and a slight decrease in *Salmonella* growth [24].

Predictive microbiology can be defined as a specialized branch of food microbiology dedicated to studying and predicting microbial behavior against environmental factors that is intrinsic to the microorganism, making use of mathematical functions for this purpose. To obtain these functions, the modeling process is carried out, which involves the use of mathematical equations that use physical and chemical laws to describe behavior [25,26,27].

An application of mathematical models to describe the inhibitory effect of UV-C treatment on *Salmonella* Enteritidis in food such as soy milk was published by Possas et al. (2018), where predictive models of inactivation were used to predict the shelf life of the product [28]. In the study, the primary and secondary models were made using computer programs such as the GInaFIT v1,6 add-in for Excel^®^. Another study, reported by Keklik et al. (2012), described the effect that this type of technology has on different microorganisms (*Salmonella* Typhimurium, *Listeria monocytogenes*, and *Salmonella* Enteritidis), concluding that the survival curves of pathogens in poultry products exposed to pulsed UV light are not linear and that the Weibull model can generally be a valuable tool to describe the inactivation patterns of pathogenic microorganisms affiliated with poultry products [29].

Therefore, to contribute to quantitative risk assessment [30], it was proposed to utilize a combination of UV light and caffeine due to the existence of previously published studies that show that a combination of various treatments is more effective than if they are applied individually. This is the case of Pagal and Gabriel (2020), where the juice was chosen as the food matrix; it was concluded that, although the heating was more effective than short-wave UV treatment, the simultaneous combined treatment of mild heat and UV-C was more effective than the combined sequential treatments tested [31].

Having obtained such positive results in previous studies with the application of UV light and caffeine jointly and individually, and taking into account the current trend of consumers towards a more natural consumption of products, this study was proposed. This study aims to determine the effect of inactivation of *Salmonella* in the chicken breast by applying a short UV light treatment and to see if, by applying a natural antimicrobial component such as caffeine, inactivation of the pathogen is also achieved. Finally, it is based on whether the two components individually manage to inhibit *Salmonella*; it is assumed that the combination of both treatments would result in their inactivation capacity being enhanced.

## 2. Materials and Methods

### 2.1. Experimental Setup

The influence of temperature on treatment (from 2 to 22 °C, with 2 °C increments) was studied to optimize the application of UV-C technology (0–15 J/cm^2^, 3 mJ/cm^2^ increment) for chicken breast decontamination in the first stage. Chicken breast samples of 20 g and 2 mm were used, and 198 samples were treated by triplicating each condition. The package was opaque to UV-C light, so the treatments needed to be carried out directly on the food matrix before the packaging.

In a second experiment, once the non-significant effect of temperature was known, a mild temperature, 14 °C, was selected for the caffeine effect study. Five caffeine concentrations (0, 5, 10, 15, and 20 nM/g) were used during UV-C treatments of chicken breast at different UV-C doses (0–15 J/cm^2^, 1 mJ/cm^2^ of increment). These conditions were set to evaluate whether the combination of UV-C irradiation with caffeine would act synergistically, resulting in an increase in *Salmonella* inactivation levels in comparison to using caffeine. This synergistic effect resulted in the reduction in the exposure time of chicken breast to UV-C radiation needed to comply with microbiological criteria. Chicken breast samples of 20 g and 2 mm were used, and a total of 240 samples were treated by triplicating each condition.

In both experiments, a total of 438 samples were analysed.

### 2.2. Chicken Breast Preparation and Characterization

Raw chicken breast meat was acquired in a local supermarket at Diamantina/MG (Brazil). The samples were transported in Styrofoam boxes to the Laboratory of Food Microbiology at the Federal University of Vales do Jequitinhonha e Mucuri for subsequent filleting and cold storage. Caffeine was purchased from Fluka Biochemika (Barcelona, Spain), stock solutions were prepared to a concentration of 20 g/L, and later, the needed concentration was spread onto a chicken breast fillet. Processing was performed at about 7 °C at the Laboratory of Food Technology with utensils previously sanitized with a solution of organic chlorine (dichlorocyanurate) (Sigma-Aldrich, Darmstadt, Germany) at a concentration of 2 g/L. 

A series of physicochemical analyses was performed for sample characterization before (control samples) and after UV-C treatments at different temperatures. Fat, protein, ash, moisture, and total acidity were determined by following AOAC methods [32]. pH was measured using a digital pH meter (PHB-500, Prolab, Diadema, Brazil). Moisture was determined using the oven drying method at 110 C for 24 h. Total protein content was determined using the Kjeldhal method. Total lipids were evaluated using the Soxhlet method.

Carbohydrates and fiber analyses were not performed on these samples since they are not relevant compounds in chicken breasts. Before inoculation, samples were kept in incubators until the temperatures set in the experimental design were achieved and stabilized. Samples were also evaluated for the presence of non-inoculated *Salmonella*. Serial decimal dilutions of UV-C-treated chicken breast samples were made in peptone water (0.9% *w*/*v*). Subsequently, 1 mL aliquots of diluted samples were pour-plated on dishes according to ISO 6579 (2017) [33] using XLD (Xilose Lisine Desoxicolate Agar, Oxoid, UK) selective agar medium. A second trial was also performed with Colorex *Salmonella* and CHROMagar *Salmonella* (Oxoid, UK) to ensure that the sample was free of countable *Salmonella*. 

### 2.3. Inoculum Preparation and Samples Inoculation

The *Salmonella enterica* subsp. *enterica* serovar Enteritidis ATCC 13076 was selected for the experiments, as it is the main serovar involved in the chicken meat contamination. The strain was obtained lyophilized from the Spanish Type Culture Collection (CECT, Valencia, Spain). After strain reconstitution, stock cultures were maintained by regular subculture on Plate Count Agar (PCA, Oxoid, UK) and stored at 4 °C. Before each experiment, a loopful of the stock culture was transferred to nutrient broth (Oxoid, UK) and incubated at 37 °C for 20 h. Then, 0.5 mL aliquots of *Salmonella* culture, previously diluted in peptone water (0.9% *w*/*v*) (Merck, Darmstadt, Germany) in order to reach a concentration of approximately 10^7^ CFU/mL, were inoculated in chicken breast samples. Afterwards, chicken breast samples were submitted to UV-C treatments. 

### 2.4. UV-C Radiation of Chicken Breast Samples

The samples were submitted to UV-C irradiation inside a lab-scale chamber made of stainless steel, which was cleaned and disinfected with 70% alcohol (*v*/*v*) before each experiment. The chamber was provided with one low-pressure mercury lamp with 9 W output, with a dominant emission wavelength of 253.7 nm. The inner surface was painted black to avoid light reflection in the walls. Chicken breast samples of 20 g and 2 mm were submitted to UV-C treatments in an inox rotator apparatus turning around an UV-C lamp. 

The apparatus was spinning, and upon reaching the opposite point in height, the sample plummeted to the bottom, resuming the process. The apparatus, being stainless steel and exposed to UV-C light, was sanitized in the section to complete a rotation before receiving the sample. The distance between the lamp and the samples was set at 10 cm. Different doses of irradiation were applied by changing the exposure time of the chicken breast samples to UV-C radiation (0–72 min), with the room temperature set to 14 °C. The UV-C doses were calculated according to Equation (1):(1)D=I×t
where *D* represents the dose of UV radiation (J/cm^2^); *I* Is the radiation intensity (mW/cm^2^); and *t* corresponds to the exposure time (s) of the product to UV light. The intensity obtained in this equipment with the configurations adopted was 3.45 mW/cm^2^. The ambient temperature was controlled with an air conditioner and monitored with an infrared thermometer (TR-300, Prolab, Brazil).

### 2.5. Microbial Analysis 

Serial decimal dilutions of UV-C-treated chicken breast samples were made in peptone water (0.9% *w*/*v*). Subsequently, 1 mL aliquots of diluted samples were pour-plated onto XLD (Xilose Lisine Desoxicolate Agar, Oxoid, Basingstoke, UK) dishes according to ISO 6579 (2017) as a selective medium for *Salmonella*. In order to decrease the detection limit, 10 mL aliquots were pour-plated onto macro XLD dishes (140 × 20 mm). The agar plates were incubated at 37 °C/24 h, and the number of survivors (CFU/mL) was determined by plate count methodology. All microbial analyses were conducted in triplicate.

### 2.6. Statistical Analysis

All the experiments were carried out on three different days to capture biological variability. The results were compared using analysis of variance (ANOVA) followed by Tukey’s test (*p* ≤ 0.05), with the software Statistica^®^ v 10 (Statsoft, Oeiras, Portugal).

### 2.7. Data Modeling

The survival curves of the test microorganism in UV-C-treated chicken breast were constructed by plotting the logarithm of the number of colony-forming units per mL of samples (log CFU/mL) against the UV-C dose (J/cm^2^). 

The GInaFiT add-in for Excel^®^ (v. 26/08/2024) [34] was applied to search for the best model. 

## 3. Results and Discussion

### 3.1. Physicochemical Characterization of the Chicken Breast 

The results of the physicochemical characterization of the chicken breast (fat, protein, ash, humidity, pH, and total acidity) are presented in Table 1. In the results obtained, no statistically significant differences were detected between the chemical composition and other physicochemical attributes determined before and after caffeine treatments at different doses (*p* > 0.05). The chemical composition of the chicken breast was consistent with other studies conducted on the effect of UV light treatment on the physicochemical characteristics of chicken breast, where minimal changes or losses of nutritional and sensory quality have been observed [13,35]. Other studies on orange juice have reported minimal changes only in pH, titratable acidity, and soluble solids [31].

### 3.2. Behavior of Salmonella enteritidis at Different Doses of UV Light in Chicken Breast at Treatment Temperatures from 2 to 22 °C

To facilitate the understanding of the data, they are represented in Figure 1, showing a clear relationship between the amount of UV light applied and the decrease in microbiological count. The data shown indicate that temperature is not an influencing factor for microbiological inhibition. It can be observed that, for an entire temperature range at a dose of UV light of 15 J/cm^2^, that is, the highest dose with which this experiment was carried out, the microbiological count did not vary. Slight changes were observed simply due to the variability of the sample. It was also observed that, when applying the non-thermal treatment, the microbiological count decreased to a minimum of 3.5 ± 0.2 CFU/g in comparison with untreated samples (7.5 ± 0.2 CFU/g). We can conclude, therefore, that as the dose of the non-thermal treatment is increased, the lethality of the microorganism increases without the temperature exerting a statistically significant effect on the result.

This behavior corresponds to that observed in other studies published in different foods treated with the same technology, such as freshly cut lettuce inoculated with *Escherichia coli* O157:H7, *Salmonella* Typhimurium, *and Listeria monocytogenes* [36]. This inactivation has also been exposed in white grape and apple juices inoculated with *Alicyclobacillus acidoterrestris* spores, showing itself as an alternative to heat treatment [37]. This non-heat treatment has also been shown to be effective on chicken breast during storage in *Campylobacter jejuni*, *Listeria monocytogenes*, and *Salmonella enterica* serovar Typhimurium, without damaging the quality of the product [15]. Positive results were also shown in *Escherichia coli* O157:H7, *Listeria monocytogenes*, *Pseudomonas aeruginosa*, and *Salmonella enterica* in liquid egg white [38].

A primary model using the adjusted equation of the relation of cell counts with time was implemented to evaluate the inactivation behavior of *Salmonella enteritidis* at different doses of UV light in chicken breasts stored at temperatures from 2 to 22 °C. The survival curves of the test microorganism in UV-treated chicken breast were constructed by plotting the logarithm of the number of colony-forming units per ml of sample (log CFU/mL) versus the short-wave UV dose (J/cm^2^) at each temperature considered (Appendix A, Table A1).

For the treatment of the data obtained which were experimentally related to microbiological growth, a complement for Microsoft © Excel, GInaFIT, was used. It was useful to test ten different types of microbial survival models in user-specific experimental data that related to the evolution of the microbial population over time. The values of the mean sum (root) of the squared errors (RMSE) and R^2^ were taken for reference, as shown in Appendix A, Table A1, and these were the models that best fit those applied in this experiment.

The Weibull model [39] (Equation (2)) was fitted to survival curves using
(2)$$\log\left(\frac{N}{N_{0}}\right)=-{\left(\frac{D}{\delta }\right)}^{p}$$
where *N* is the number of survival cells (log CFU/mL) after UV-C treatments; *N*_0_ is the number of cells before UV-C treatments (log CFU/mL); *D* is the UV-C dose (J/cm^2^); *p* is the model shape parameter (dimensionless); and *δ* is the model scale parameter, representing the dose required to achieve the first tenfold reduction in the population (J/cm^2^). 

The inactivation rate for the Weibull model was (Kmax = 0.70–0.76) for the range of all temperatures studied in the highest UV dose (15 J/cm^2^).

This inactivation model has also been obtained in other studies related to *Salmonella* Typhimurium in dry fermented sausages, to which short-wave UV light has been applied [40]. On the other hand, this model has served to evaluate the inactivation effect using the same non-thermal treatment in *Salmonella* Enteritidis in a soymilk matrix [28]. It has also been shown that *Escherichia coli* O157:H7 and *Listeria monocytogenes* inoculated in apple juice show this inactivation pattern when an isothermal treatment is applied by microwave heating [41].

It can also reach a good fit with the biphasic model [42], as represented in Equation (3). These values were adjusted to the survival curves using the GInaFIT add-in for Excel^®^ [34], with the best fit being the biphasic model.

〖Log〗_10 N =〖Log〗_10 N_0 ±〖Log〗_10 (f·e^((−〖Kmax〗_1·t)) ± (1-f)· e^((−〖Kmax〗_2·t)))
(3)

where N is the number of surviving cells (log CFU/mL) after UV-C treatments; N_0_ is the number of cells before UV-C treatments (log CFU/mL); f is the fraction of the initial main subpopulation; t corresponds to the time of exposure of the product to UV light; and Kmax is the constant of the rate of inactivation of the first order (1/time unit), where Kmax^1^ (1.38–1.99) and Kmax^2^ (0.33–0.40) represent the rate of decrease for initially larger and smaller populations (constant, after the shoulder and/or before the tail).

This inactivation model has also achieve a good fit in other food matrices, such as dry persimmon, with fungi such as *Cladosporuim* spp., *Aspergillus* spp., *Penicillium* spp., and *Alternaria* spp., among others, showing that this technology is promising for food surface decontamination [43].

Other studies performed on chicken breast with UV light treatment at doses ranging from 0 to 0.5 J/cm^2^ resulted in reductions of no more than 1.3 log CFU/g of *Salmonella enterica* serovar Typhimurium [15]. Treatment of 0.192 J/cm^2^ caused a reduction of up to 1.3 and 4.2 log10 CFU/cm^2^ on serovar Enteritidis [44].

### 3.3. Primary Model of the Inactivation of Salmonella enteritidis at Different Doses of Caffeine in Chicken Breast at a Constant Temperature of 14 °C

We decided to carry out this study at a temperature close to refrigeration (14 °C), where the risk of *Salmonella* on fresh chicken breasts is high. An analysis of variance (ANOVA) of the effect of different concentrations of caffeine was executed with the addition of Tuckey’s post hoc test (*p* ≤ 0.05), again obtaining non-significant values.

As previously described by Maletta and Were (2012) [24], coffee filtering by itself did not present a significant antimicrobial effect in chicken breast, although in our case, there was a correlation between the presence of coffee and a slight decrease in *Salmonella* growth, as can be seen in Table 2. The best result of pathogen reduction (1.5 log CFU/g) occurred with the highest caffeine concentration of 20 nM/g. In addition, caffeine has been shown to continue to be effective in its inhibition against *Salmonella* depending on the dose used for the two *Salmonella* serotypes, both one day after storage and during storage, presenting growth delays or total inhibition [24]. In the same line, the antimicrobial effect against *E. coli* O157:H7 has been demonstrated in liquid medium and skim milk [45,46]. Other studies have demonstrated this antibacterial effect of Arabica coffee extracts against *Streptococcus mutans* [46].

When modeling the data through the GInaFIT program, based on R^2^ and RMSE values (Appendix A Table A3), the model that showed the best fit was the linear logarithmic regression model [47], for which Equation (4) shows the identified inactivation model:(4)$$N=N_{0}\cdot e^{-K_{max}\cdot t}$$
where N is the number of survival cells (log CFU/mL) after caffeine treatment; N_0_ is the number of cells before the treatments (log CFU/mL); t corresponds to the product’s exposure time to UV light; and K_max_ (0.17 ± 0.01 in this model) is the first-order inactivation rate constant (1/time unit).

Caffeine is one of the active pharmaceutical ingredients (API) most widely studied for co-crystallization. It forms caffeine co-crystals with organic substrates [48] and with different acids, such as maleic acid [49], oxalic acid [50], and malic and mesaconic acid [51], among others. The antibacterial activity showed that the co-crystal ([(1,10-PhenH^±^) (caf) (${PF}_{6}^{-}$)]) had better activity against the following bacteria: *Escherichia coli*, *Staphylococcus aureus*, *Klebsiella pneumonia*, *Klebsiella oxytoca,* and *Pseudomonas putida* [52].

The experimental data exposed herein show that a dose of caffeine greater than 15.0 nM/g decreases the rate of inactivation of the microorganism (Table 2), so the substance would be wasted. The optimum point of application is 15.0 nM/g.

### 3.4. Inactivation Effect of UV and Caffein on Salmonella enteritidis 

Regarding the statistical analysis, a battery of different tests was carried out to better describe the influence of caffeine and/or UV-C radiation on bacterial growth. Thus, we started from the premise of knowing the “influence” of these factors separately. Later, we proceeded to analyze the effect of both treatments together on the mentioned growth. 

A linear regression of these factors against bacterial growth was carried out without finding clear significance. For all the tests, the SPSS software S. (2019, v 11) was used (SPSS for Windows Release, 25). This prevented us from making a secondary model that would integrate the set of caffeine doses, UV light, and temperature range.

The results obtained after the combined treatment of UV-C and caffeine (Table 3) showed that apparently, they are influential factors on the microbiological count, although statistical analysis indicated no significant differences. 

On the other hand, it was observed that when caffeine was applied (Figure 2), the microbiological count decreased. If the microbiological count obtained in a sample that had not been subjected to UV light was observed, this was 7.5 ± 0.2 CFU/g. When the natural inhibiting substance was applied, this value decreased to a minimum of 6.0 ± 0.2 CFU/g. We can conclude, therefore, that as the dose of this substance increases, the lethality of the microorganism increases.

When modeling the data (Appendix A Table A3), the double Weibull model (Equation (5)) was the best adjusted to the survival curves. We used the GInaFiT add-in for Excel^®^:(5)$$N=\frac{N_{0}}{\left(1\pm 10^{\alpha }\right)}\cdot \left[10^{{-\left(\frac{t-1}{\delta_{1}}\right)}^{p_{1}}\pm \alpha }\pm 10^{{-\left(\frac{-1}{\delta_{2}}\right)}^{p_{2}}}\right]$$
where N is the number of surviving cells (log CFU/mL) after UV-C treatments; N_0_ is the number of cells before UV-C treatments (log CFU/mL); t corresponds to the time of exposure of the product to UV light; α is the fraction of the first remaining subpopulation in the total population and is defined as the logarithm of f and is equivalent to α = log10 (N01/N02); p is the shape parameter of the model (dimensionless); and δ is the scale parameter of the model, representing the dose required to achieve the first ten-fold reduction in the population (J/cm^2^).

The Weibull model + tail [39] (Equation (6)) was fitted to the survival curves using the GInaFiT add-in for Excel^®^ [34]:(6)$$N={(N}_{0}-N_{res})\cdot \;10^{\left({-\left(\frac{t}{\delta }\right)}^{p}\right)\;}+N_{res}$$
where N is the number of surviving cells (log CFU/mL) after UV-C treatments; N_0_ is the number of cells before UV-C treatments (log CFU/mL); N_res_ is the starting point of the tail (log CFU/mL); t corresponds to the time of exposure of the product to UV light; p is the shape parameter of the model (dimensionless); and δ is the scale parameter of the model, representing the dose required to achieve the first ten-fold reduction in the population (J/cm^2^).

It can also reach a good fit with the biphasic model [42], as represented in Figure 3, where the red curve is the fitted model and the blue diamond markers are the data observed in the study and represented in Equation (3), where these values were adjusted to the survival curves using the GInaFIT add-in for Excel^®^ [34]. The best fit was the biphasic model, Equation (3).

Observing the results obtained, it was affirmed that there was a greater reduction in the microbiological count when applying both non-thermal treatments simultaneously than individually but sequentially, with the optimal application points being 15.0 nM/g and 7.0 J/cm^2^. 

In summary, all the conditions previously described had a slight influence on *Salmonella*, but after performing the statistical analysis, it was found that this influence was not significant, so it is not possible to fit our data into a secondary model.

In addition, the models are often compared with data from previously published studies for validation using specific statistical methodologies. However, due to the lack of data in the consulted bibliography, the model could not be validated with the described precision and bias factors [53].

## 4. Conclusions

Finally, it can be concluded that the influence of caffeine and UV radiation on the inactivation of *Salmonella* enteritidis in fresh chicken breast is very promising when applied in combination, leading to reductions in the pathogen load of up to 6.3 log CFU/g. On the other hand, it was observed that, in samples submitted only to caffeine, the greatest reduction was 1.5 log CFU/g in the highest dose, and in the case of UV light, this was 4.0 CFU/g. By applying the predictive models, a good fit was achieved in each of the cases studied, reaching R^2^ values from 0.9998 to 0.9553. The most effective combination for the reduction in *Salmonella* was 12 J/cm^2^ of UV light and 15 nM/g of caffeine.

## Figures and Tables

**Figure 1 microorganisms-12-01805-f001:**
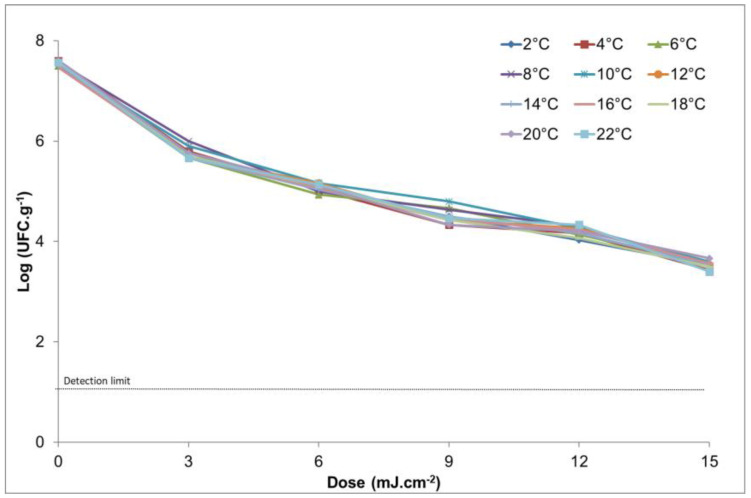
Graphic representation of the microbiological count (log CFU/g) when applying different doses of UV light (0, 3, 6, 9, 12, 15 J/cm^2^) on the reduction in *Salmonella* at different temperatures.

**Figure 2 microorganisms-12-01805-f002:**
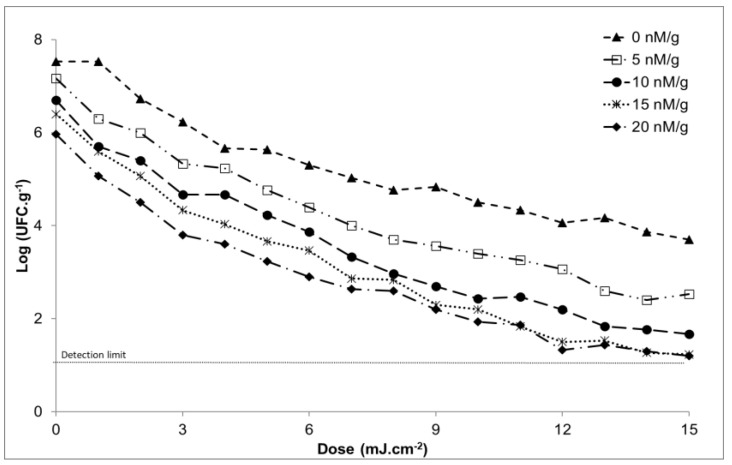
Graph that represents doses of UV light vs. log at different doses of caffeine at a temperature of 14 °C.

**Figure 3 microorganisms-12-01805-f003:**
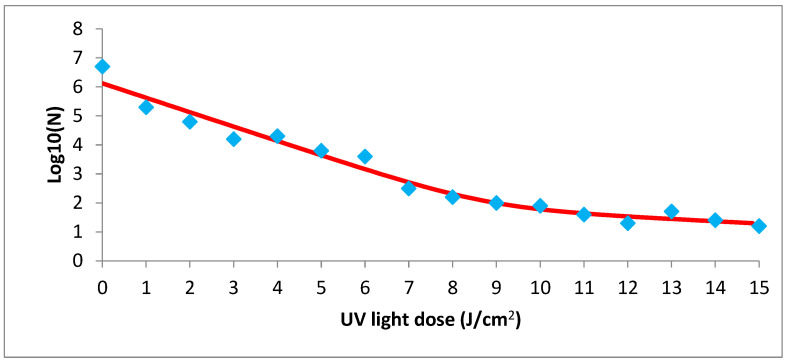
Fit of observed data (blue squares) for the biphasic model of the treatment of UV light with 15 nM/g of caffeine.

**Table 1 microorganisms-12-01805-t001:** Physical–chemical characterization (average ± SD) of the chicken breast after the application of caffeine at doses of 0, 5, 10, 15, and 20 nM/g.

Caffeine (nM/g)	Protein (%)	Fat (%)	Humidity (%)	Ash (%)	Total Acidity (%)	pH	Absorption Coefficient (cm^−1^)
0	20.82 ± 0.18	2.85 ± 0.08	74.85 ± 2.28	1.69 ± 0.05	0.21 ± 0.05	5.87 ± 0.08	959.2 ± 46.7
5	20.97 ± 1.48	2.84 ± 0.07	76.56 ± 2.40	1.66 ± 0.10	0.24 ± 0.05	5.84 ± 0.06	961.5 ± 31.2
10	21.13 ± 2.00	2.86 ± 0.12	75.40 ± 2.18	1.67 ± 0.06	0.22 ± 0.03	5.82 ± 0.08	965.8 ± 42.9
15	20.77 ± 1.07	2.80 ± 0.09	76.69 ± 1.70	1.65 ± 0.08	0.23 ± 0.03	5.84 ± 0.05	961.3 ± 79.7
20	21.18 ± 1.70	2.86 ± 0.05	74.94 ± 2.71	1.67 ± 0.04	0.24 ± 0.02	5.82 ± 0.12	962.6 ± 49.1

**Table 2 microorganisms-12-01805-t002:** Effect of caffeine microbiological counts (log CFU/g) of *Salmonella* in chicken breast at a constant temperature of 14 °C and the inactivation rate.

Caffeine Dose (nM/g)	Without Caffeine *	With Caffeine *	Inactivation Rate
0	7.5 ± 0.2	7.5 ± 0.2	0.23
5	7.6 ± 0.2	7.2 ± 0.0	0.30
10	7.6 ± 0.2	6.7 ± 0.1	0.32
15	7.6 ± 0.2	6.4 ± 0.2	0.33
20	7.5 ± 0.2	6.0 ± 0.2	0.29

* Average log (CFU/g) ± SD log (CFU/g).

**Table 3 microorganisms-12-01805-t003:** Microbiological counts (CFU/g) of *Salmonella* in chicken breast subjected to different doses of UV light in combination with different doses of caffeine at 14 °C.

	Caffeine (nM/g)				
Dose (J/cm^2^)	0	5	10	15	20
0	7.5 ± 0.2	7.2 ± 0.0	6.7 ± 0.1	6.4 ± 0.2	6.0 ± 0.2
1	6.7 ± 0.2	6.3 ± 0.1	5.7 ± 0.1	5.6 ± 0.2	5.1 ± 0.2
2	6.2 ± 0.2	6.0 ± 0.1	5.4 ± 0.1	5.1 ± 0.2	4.5 ± 0.3
3	5.7 ± 0.1	5.3 ± 0.2	4.7 ± 0.2	4.3 ± 0.1	3.8 ± 0.2
4	5.6 ± 0.2	5.2 ± 0.2	4.7 ± 0.2	4.0 ± 0.2	3.6 ± 0.2
5	5.3 ± 0.1	4.8 ± 0.1	4.2 ± 0.2	3.7 ± 0.1	3.2 ± 0.2
6	5.0 ± 0.0	4.4 ± 0.1	3.9 ± 0.0	3.5 ± 0.1	2.9 ± 0.3
7	4.8 ± 0.3	4.0 ± 0.1	3.3 ± 0.2	2.9 ± 0.2	2.6 ± 0.3
8	4.8 ± 0.2	3.7 ± 0.3	3.0 ± 0.2	2.8 ± 0.6	2.6 ± 0.4
9	4.5 ± 0.1	3.6 ± 0.4	2.7 ± 0.5	2.3 ± 0.2	2.2 ± 0.1
10	4.3 ± 0.3	3.4 ± 0.2	2.4 ± 0.4	2.2 ± 0.2	1.9 ± 0.2
11	4.1 ± 0.3	3.3 ± 0.3	2.5 ± 0.5	1.8 ± 0.2	1.9 ± 0.4
12	4.2 ± 0.5	3.1 ± 0.3	2.2 ± 0.2	1.5 ± 0.1	1.3 ± 0.2
13	3.9 ± 0.4	2.6 ± 0.4	1.8 ± 0.2	1.5 ± 0.4	1.4 ± 0.3
14	3.7 ± 0.2	2.4 ± 0.2	1.8 ± 0.1	1.3 ± 0.2	1.3 ± 0.3
15	3.5 ± 0.5	2.5 ± 0.4	1.7 ± 0.3	1.2 ± 0.2	1.2 ± 0.2

Mean log (CFU/g) ± SD log (CFU/g).

## Data Availability

The raw data supporting the conclusions of this article will be made available by the authors on request.

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
