# Peer review of "Salmonella Inactivation Model by UV-C Light Treatment in Chicken Breast"

_microorganisms, 2024, doi:10.3390/microorganisms12091805_

Round 1

Reviewer 1 Report

Comments and Suggestions for Authors

The article submitted to microorganisms, "Salmonella Inactivation Model by UV-C Light Treatment in Chicken Breast," is an interesting article dealing with the potential of UV-C light to reduce the contamination of chicken with Salmonella. The authors studied three factors: the UV light treatment, the temperature, and the amount of caffeine (that was not outlined in the title, probably because it is a kind of sub-study).

Considering that Salmonella continues to be one of the main concerns in foodborne diseases and the enormous importance that chicken has in this context, any attempt to contribute to building solutions to reduce this problem is welcome.

The main concern I found in this work was in the MM section, where it was always obvious to the reader what was really done. Some details are only known when the reader reaches the results section. Due to this issue, the article seems to be poorer than it really is in some parts.

I kindly suggest that the authors make a detailed revision of the MM section to show clearly all the work done and how it was done in that section of the article.

Please ensure that the three factors were not studied in a full-factorial experiment and that the caffeine study was made apart."

English is not my natural language. Considering that, I found the language adequate for an international journal.

There are some detailed notes below.

Line 21. The phrase “The temperatures used were from 2 to 22°C. is not completely clear once the caffeine experiments were done only at 14ºC. The current writing suggests that all combinations of all factors were tested. This issue must be clarified in the abstract.

Line 23. Weibull + cola might it be weibulk+tail (lost in translation from Spanish)

Line 130. 2 to 22ºC – with which increments (only when the reader arrives to figure 1 it is known that is 2ºC of increment)

Line 133 - 0-15 J/cm2 – with which increments (only when the reader arrives to figure 1 it is known that is 3mJ.cm2 of increment)

Lines 146-148. In this kind of experiment, contamination prevention (description of the work clothes!) is expected. This information is superfluous. Please consider reducing it to a simpler note.

Line 149. Somewhere in this part of the MM sections, authors should refer to the size of the samples and how many steaks were processed (when all the temperatures and UV conditions are known, it would be easier to infer the dimension/robustness of the work being evaluated to be published. Were there repetitions, or was it made without repetitions?

Line 150 and linked parts of the work. What was the hypothesis underlying the need to analyze the breasts with different amounts of caffeine and UV treatments? If it is only to characterize the material being used, and once any variation observed was most probably due to the natural variability among individual chickens, please consider presenting the results not in the table for each caffeine amount but in the text as the mean and SD for all (unless there is an argument to believe that caffeine influences the chemical composition!).

Line 151—The authors use "Fat" in the MM section (and I agree with that designation). In Table 1, it is referred to as "grease." Please uniformize the language.

Line 155 – 157. After referring to AOAC methods, the authors briefly referenced the methods in this text segment. If the authors consider it important to make that note, please add a note on total acidity.

Line 158-159. I suggest deleting this phrase. I do not see the need for this justification.

Line 166 – “free of Salmonella”. Considering the procedures used to search for the presence of Salmonella in foods with several recovering steps, I would prefer to refer to this procedure as “free of countable salmonella”

Line 168-170. This phrase is missing something. "as the contamination" is not resolved in the following text. I believe that the authors meant "once it is the main serovar involved in the chicken meat contamination."

Line 176-177. “reach a concentration of approximately 107 CFU/mL Is this the concentration of the suspension to add to the meat, or is it already the expected concentration in the meat? If the second case is correct, please check the units for CFU/g or CFU/cm2.

Line 178. Just to confirm: samples were not packaged before UV.C exposure? Do the authors defend that procedure for industry application? What might be the difficulties introduced by the package? Please consider adding some information somewhere in the article on this subject that might interest some readers.

Line 184 – 185. “The sample was homogenized is repeated with the next phrase.

Line 185. The sample was homogenized by tipping. Was it the sample or the UV exposure that was homogenized?

MM general doubt. Where/when were the samples subjected to temperatures between 2 and 18ºC? Was it before the UV treatment or during the UV treatment? In line 265, it is indicated that the temperature of storage (storage before or after the treatment? How long was that storage?

Figure 1 and Table 2 have the same data. I do not know exactly the policy of this journal, but it is usually recommended to not repeat results in graphs and tables.

Author Response

The article submitted to microorganisms, "Salmonella Inactivation Model by UV-C Light Treatment in Chicken Breast," is an interesting article dealing with the potential of UV-C light to reduce the contamination of chicken with Salmonella. The authors studied three factors: the UV light treatment, the temperature, and the amount of caffeine (that was not outlined in the title, probably because it is a kind of sub-study).

Considering that Salmonella continues to be one of the main concerns in foodborne diseases and the enormous importance that chicken has in this context, any attempt to contribute to building solutions to reduce this problem is welcome.

The main concern I found in this work was in the MM section, where it was always obvious to the reader what was really done. Some details are only known when the reader reaches the results section. Due to this issue, the article seems to be poorer than it really is in some parts.

I kindly suggest that the authors make a detailed revision of the MM section to show clearly all the work done and how it was done in that section of the article.

Please ensure that the three factors were not studied in a full-factorial experiment and that the caffeine study was made “apart."

English is not my natural language. Considering that, I found the language adequate for an international journal.

Response: The authors thank the reviewer for his/her positive feedback about the quality of the paper and have improved the wording to make it easier to understand

Two factors were studied in UV-C treatment: temperature and caffeine, and no synergistic effect was observed. This is why the authors consider the title initially proposed to be more adequate.

The authors detailed the M&M section to clearly show all the work done. Three factors were studied, but not in a full-factorial experiment, as has been clarified in the M&M section.

There are some detailed notes below.

Line 21. The phrase “The temperatures used were from 2 to 22°C.” is not completely clear once the caffeine experiments were done only at 14ºC. The current writing suggests that all combinations of all factors were tested. This issue must be clarified in the abstract.

Response: This issue has been clarified in the abstract text.

Line 23. Weibull + cola – might it be weibulk+tail (lost in translation from Spanish…)

Response: It has been corrected in the text according to the reviewer's recommendation.

Line 130. 2 to 22ºC – with which increments (only when the reader arrives to figure 1 it is known that is 2ºC of increment)

Response: It has been corrected in the text according to the reviewer's recommendation.

Line 133 - 0-15 J/cm2 – with which increments (only when the reader arrives to figure 1 it is known that is 3mJ.cm2 of increment)

Response: It has been corrected in the text according to the reviewer's recommendation.

Lines 146-148. In this kind of experiment, contamination prevention (description of the work clothes!) is expected. This information is superfluous. Please consider reducing it to a simpler note.

Response: The text has been simplified according to the reviewer's recommendation.

Line 149. Somewhere in this part of the MM sections, authors should refer to the size of the samples and how many steaks were processed (when all the temperatures and UV conditions are known, it would be easier to infer the dimension/robustness of the work being evaluated to be published. Were there repetitions, or was it made without repetitions?

Response: The M&M section has been improved as suggested by reviewer:

“The influence of temperature on treatment (from 2 to 22 ° C, with 2ºC increment) was studied to optimize the application of UV-C technology (0-15 J/cm2, 3 mJ/cm2 increment) for chicken breast decontamination in the first stage. Chicken breast samples of 20g and 2mm were the sample's size, and 198 samples were treated by triplicating each condition. The package is opaque to the UV-C light, so the treatments must be done directly on the food matrix, before the packaging.

In a second experiment, and once known the non-significant effect of temperature, it was selected a mild temperature, 14ºC for the caffeine effect study. Five caffeine con-centrations (0, 5, 10, 15 and 20 nM/g) were used during chicken breast UV-C treatments at different UV-C doses (0-15 J/cm2, 1mJ/cm2 of increment). These conditions were set to evaluate if the combination of UV-C irradiation with caffeine would act synergistically, resulting in an increase of Salmonella inactivation levels in comparison to using caffeine. This synergistic effect would result in the reduction of the exposure time of chicken breast to UV-C radiation needed to comply with microbiological criteria. Chicken breast samples of 20g and 2mm were the size of the sample, and a total of 240 samples were treated by triplicating each condition.

In both experiments, a total of 438 samples were analyzed.”

Line 150 and linked parts of the work. What was the hypothesis underlying the need to analyze the breasts with different amounts of caffeine and UV treatments? If it is only to characterize the material being used, and once any variation observed was most probably due to the natural variability among individual chickens, please consider presenting the results not in the table for each caffeine amount but in the text as the mean and SD for all (unless there is an argument to believe that caffeine influences the chemical composition!).

Although it is true that the compositional variation depending on the dose of caffeine (Table 1) is minimal, we consider that this is relevant information that may be of interest to readers and should be kept in the work.

Line 151—The authors use "Fat" in the MM section (and I agree with that designation). In Table 1, it is referred to as "grease." Please uniformize the language.

Response: The term was corrected in the Table as suggested by reviewer

Line 155 – 157. After referring to AOAC methods, the authors briefly referenced the methods in this text segment. If the authors consider it important to make that note, please add a note on total acidity.

Response: Total acidity was performed according to AOAC, being contemplated by the phrase in the original text. (AOAC 20.118 )

Line 158-159. I suggest deleting this phrase. I do not see the need for this justification.

Response: The text was eliminated as suggested by reviewer.

Line 166 – “free of Salmonella”. Considering the procedures used to search for the presence of Salmonella in foods with several recovering steps, I would prefer to refer to this procedure as “free of countable Salmonella

Response: The suggestion was incorporated in the text.

Line 168-170. This phrase is missing something. "as the contamination" is not resolved in the following text. I believe that the authors meant "once it is the main serovar involved in the chicken meat contamination."

Response: The suggestion was incorporated in the text.

Line 176-177. “reach a concentration of approximately 107 CFU/mL” Is this the concentration of the suspension to add to the meat, or is it already the expected concentration in the meat? If the second case is correct, please check the units for CFU/g or CFU/cm2.

Response: It is referred to the suspension of the inoculum. The text in the manuscript has been corrected.

Line 178. Just to confirm: samples were not packaged before UV.C exposure? Do the authors defend that procedure for industry application? What might be the difficulties introduced by the package? Please consider adding some information somewhere in the article on this subject that might interest some readers.

Response: The package is opaque to UV-C light, so the treatments must be done directly on the food matrix before the packaging. Samples were not packaged for treatment as the effect is on the surface of the fillet. Food industries may package it afterward.

Line 184 – 185. “The sample was homogenized” is repeated with the next phrase.

Response: The text was eliminated as suggested by reviewer.

Line 185. The sample was homogenized by tipping. Was it the sample or the UV exposure that was homogenized?

Response: The sample was inserted on a rotating cylinder around an UV-C lamp, being the sample exposed on both sides by tipping.

It has been corrected in the text.

MM general doubt. Where/when were the samples subjected to temperatures between 2 and 18ºC? Was it before the UV treatment or during the UV treatment? In line 265, it is indicated that the temperature of storage (storage before or after the treatment? How long was that storage?

Response: The samples were fresh and immediately treated in the chambers at the indicated temperatures. The samples were treated at different temperatures, looking at the optimization of the process. The storage of samples was not studied, only the inactivation of the pathogen.

Figure 1 and Table 2 have the same data. I do not know exactly the policy of this journal, but it is usually recommended to not repeat results in graphs and tables.

Response: The table was eliminated as suggested.

Reviewer 2 Report

Comments and Suggestions for Authors

The authors present an assessment of Salmonella inactivation on chicken with a combination of caffeine and UV. the work is similar to their previously published study, which was focused on E. coli inactivation on the same matrix with the same treatments - https://doi.org/10.1016/j.foodcont.2021.108206

1. as this study is very similar to the previous one, it is odd that it is not mentioned in the introduction as supporting information for the current study. 

2. lines 69-71 - these two things don't align. how does microbial inactivation minimize the loss of nutritional quality?

3. lines 89-95 - this is not the correct definition of QMRA. "(QMRA) is a mathematical modeling approach used to estimate the risk of infection and illness when a population is exposed to microorganisms in the environment." the focus is on the change in risk of illness. this needs to be modified, or could even be removed.

4. lines 186 -192 - this process needs to be better described. what is mean by 'tipping'? it is difficult to understand the dose applied based on this description.

5. table 2 and 4 - would be useful to include statistical comparisons.

Comments on the Quality of English Language

- grammatical issues and typos throughout.

- proper italicization of Salmonella throughout

- proper formatting for Salmonella serovars - not italicized but capitalized

- unclear word choices in many places. for example, line 186 - should this be rotating rather than tipping? line 235 and the legend to figure 1 - what is mean by 'development' here? microbial growth, or inactivation?

Author Response

The authors present an assessment of Salmonella inactivation on chicken with a combination of caffeine and UV. the work is similar to their previously published study, which was focused on E. coli inactivation on the same matrix with the same treatments - https://doi.org/10.1016/j.foodcont.2021.108206.

as this study is very similar to the previous one, it is odd that it is not mentioned in the introduction as supporting information for the current study. 

Response: This reference is cited in manuscript, number 21

Lines 69-71: These two things don't align. How does microbial inactivation minimize the loss of nutritional quality?

Response: Bacterial activity can increase food degradation and affect its nutritional quality. The text refers to UV treatment that will not affect nutritional quality.

lines 89-95 - this is not the correct definition of QMRA. "(QMRA) is a mathematical modeling approach used to estimate the risk of infection and illness when a population is exposed to microorganisms in the environment." the focus is on the change in risk of illness. this needs to be modified, or could even be removed.

Response: The text has been removed as suggested.

lines 186 -192 - this process needs to be better described. what is mean by 'tipping'? it is difficult to understand the dose applied based on this description.

Response: The sample was inserted on a rotating cylinder around an UV-C lamp, being the sample exposed on both sides by tipping. This mistake has been corrected in the text.

  1. table 2 and 4 - would be useful to include statistical comparisons.

Response: Table 2 was eliminated. The information in these two tables is discussed in the R&D section.

- grammatical issues and typos throughout, and  proper italicization of Salmonella throughout

Response: All the text was checked and corrected for English grammatical issues and proper italicization of bacteria names.

- proper formatting for Salmonella serovars - not italicized but capitalized

Response: All the text was checked and corrected.

- unclear word choices in many places. for example, line 186 - should this be rotating rather than tipping?

Response: The sample was inserted into a rotating cylinder around a UV-C lamp, and it was exposed on both sides by tipping. This mistake has been corrected in the text.

 line 235 and the legend to figure 1 - what is mean by 'development' here? microbial growth, or inactivation?

Response: This terms is referred to “count”, it has been corrected all around the manuscript.
